# How fast-and-frugal trees can inform diagnostic and intervention decisions for enhancing elite athlete performance

Lena Siebert[1]*, Lukas Reichert[1], Lisa Musculus[2], Laura Will[2], Ahmed Al-Ghezi[3], Markus Raab[2], Karen Zentgraf[1]

**1** Movement and Exercise Science, Institute of Sports Sciences, Goethe University Frankfurt, Frankfurt am Main, Germany, **2** Department of Performance Psychology, Institute of Psychology, German Sport University Cologne, Cologne, Germany, **3** Institute of Computer Science, Goethe University Frankfurt, Frankfurt am Main, Germany

* siebert@sport.uni-frankfurt.de

## Abstract

The key to fostering the individual potential of an elite athlete lies in deciding what to prioritize in training. Heuristic decision tools such as fast-and-frugal trees (FFTrees) have proven to be effective and suitable for identifying promising determinants in this context. FFTrees are binary decision trees that can make decisions based on only one reason. The objective of this study was to examine the applicability of FFTrees to inform intervention decisions in elite athletes. We aimed to create FFTrees and evaluate their ability to determine individually beneficial interventions. We collected cognitive, psychosocial, and motor-performance diagnostic data from 466 German elite athletes in different sports disciplines. First, we used principal component analysis to identify components representing types of interventions across sports. These served as cues for the FFTrees. As a result, the PCA identified six cues. Two sport-specific FFTrees were created using these six cues. One FFTree was created for trampoline with four cues (relative grip strength, motor cost, motor inhibition, visual selective attention) and 90% correct predictions. The other FFTree was created for volleyball with four cues (motor inhibition, motor cost, countermovement jump, Y-Balance Test) and 75% correct predictions. To conclude, the high accuracy confirms that FFTrees enable data-based decisions for interventions based on sport-specific demands and the preferences of coaches. We argue that FFTrees are beneficial in projects collecting multidisciplinary variables for personalizing interventions in elite athletes. Coaching practice benefits from using FFTrees by providing reference values when an intervention could enhance performance. In the future, we advocate that team sports develop position-specific FFTrees. In conclusion, FFTrees empower decision-makers by efficiently identifying athletes' adaptation potentials.

**Data availability statement:** Data for the creation of the FFTrees are publicly shared. Data are available here https://doi.org/10.17605/OSF.IO/KFRC2. The code underpinning our findings can be accessed through the R Package developed by Phillips et al. [42]. The latest developer version of FFTrees is available at http://www.github.com/ndphillips/FFTrees. The code and documentation presented here were created with the latest FFTree version 2.0.0. The specific formula used to create the FFTrees in this study is provided in the caption of the Supporting Information S 4. The data analyzed for the purposes of this study were part of a multidisciplinary large-scale data set that included multiple points of measurement and a cross-sectional and longitudinal perspective. The subset of data included in this study covered the cross-sectional data collected in the period 01/2022–12/2023.

**Funding:** This work was supported by the Bundesinstitut für Sportwissenschaft (German Federal Institute of Sport Science, https://www.bisp.de) in 2021–2025 under Grant No. 081901/21-25 (English title: Individual performance development in elite sports by holistic and transdisciplinary process optimization). Specifically, LR, LM, LW, AA received funding. Principal investigators were KZ, MR. The funders had no role in study design, data collection and analysis, decision to publish, or preparation of the manuscript.

**Competing interests:** The authors report there are no competing interests to declare.

# 1 Introduction

One essential goal in elite sports is to improve athletes in terms of their health and their performance-related determinants. Identifying and developing those who will become the next Olympic champions is an aspiration of most of the stakeholders involved in elite sports, from athletes and coaches to sports associations and countries. Hence, developing methods to improve performance is a fundamental task. Recent findings in sports science suggest that athletes should be viewed as complex individuals who differ in various person- and situation-related factors [1]. Being able to account for such interindividual variance has motivated stakeholders in both the applied field and research to acquire rich datasets on athletes. Among the many possible ways to foster the individual potential of elite athletes, the key to improving individual performance lies in deciding which determinants to prioritize and to translate into systematic training interventions.

However, analyzing data adequately and deriving interventions from performance-related variables can be difficult. Therefore, we argue that heuristic decision tools can be used to reduce the amount of information to the most valid aspects. In this article, we show how fast-and-frugal trees (FFTrees) can be effective decision tools with which to reach data-driven choices, such as deciding upon suitable training interventions in elite sports.

To enhance athletes' performance, coaches and sports scientists need to adopt an individualized approach to training. Studies have demonstrated how responses to identical stimuli as well as performance determinants (e.g., $VO_{2max}$) can differ between athletes [2,3]. It has been argued that, alongside genetic influences, sociological, psychological, and epigenetic determinants also contribute to these interindividual adaptations [4]. Sigmund and Güllich [5] showed that individualization of training was associated with improved long-term development of performance. In practice, however, individualization is implemented only in part, and mostly through athlete monitoring and the management of external versus internal load. However, to meet the high competitive demands, individualization becomes relevant in decisions on training interventions [6]. Nonetheless, not every intervention is equally effective in improving individual performance. Therefore, it is important to select appropriate interventions on an individual basis. To determine where athletes show their greatest individual adaptation potential, decisions should be informed by data that encompass performance-related variables. Selecting the necessary diagnostics and relevant determinants for this can be guided by theoretical models of expertise.

Recent research on sporting talent and expertise indicates that sports expertise is multifaceted. One recent model has described expertise as an interaction between genetic and environmental factors that impacts several additional determinants of expertise [7]. Building on this gene–environment interaction model, Zentgraf and Raab [8] situated the model in the context of elite sports and demonstrated how this understanding of sports expertise can inform sports-science diagnostics. They determined that diagnostics should consist of a holistic, multidisciplinary approach including physiological, psychological, psychosociological, and motor-performance assessments (Fig 1).

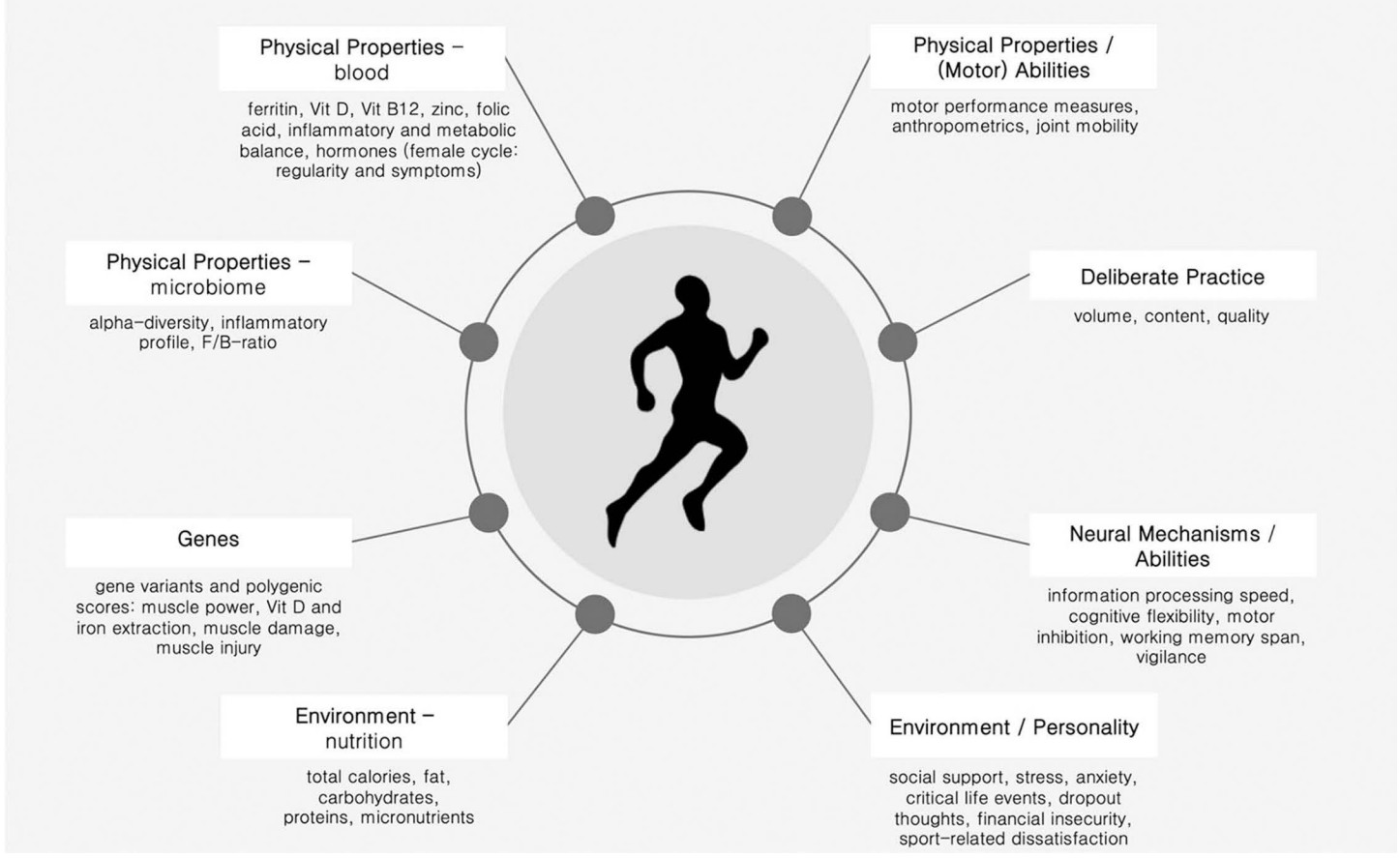

**Fig 1. Diagnostics assessed based on the model of Ullén et al. [7] adapted in Zentgraf and Raab [8].** F/B = Firmicutes/Bacteroidetes, Vit = Vitamin.

When implementing such multidisciplinary diagnostics in elite sports, Zentgraf et al. [1] found several variables associated with sports expertise that reinforced the need to consider multidisciplinary domains when deriving interventions for elite athletes.

In order to identify individually tailored training interventions, these findings indicate three main challenges: first, selecting the most impactful determinant among numerous determinants influencing athletic performance; second, deciding on an intervention with the most promising individual adaptation potential constrained by the time available in an athlete's training schedule; and third, handling the large datasets generated by accommodating the multidisciplinary approach to diagnostics and the individualized analysis. It is not feasible to implement all the indicated interventions. Instead, an informed decision is needed to enable the selection of the intervention with the greatest potential for adaptation for individual athletes and the lowest risk of showing limited effects [5,6]. These challenges are aggravated by the absence of norm data, because most measures have been validated in the general population and may not be fully applicable to athletes. Thus, when coaches use previous performance and diagnostic data to decide on team or individual training regimes, choosing between multiple training options is accompanied by uncertainty due to multiple potential outcomes of a specific intervention. This makes it difficult for stakeholders to make accurate decisions or predictions [9].

One methodological solution for dealing with decision-making characterized by uncertainty is to use FFTrees. These can be used to manage the constraints associated with data captured in elite sports without any need to engage in complex computation

   

[10]. This ensures that they are simple for coaches and sports scientists to implement. We propose FFTrees as a suitable tool for overcoming the above-mentioned challenges facing research and practice when aiming to develop, implement, and evaluate effective training interventions in elite sports. FFTrees are suitable from a conceptual perspective: As part of the heuristic family, they are grounded in Simon's [11] theories of bounded rationality. Bounded rationality conceptualizes human rationality as limited by information-processing capacities and shaped by environmental constraints. In real-life situations, decision making is often influenced by uncertainty, limited information, and limited processing capacities. This is in contrast to the assumptions of unbounded rationality, in which knowledge of all probabilities, consequences, and alternatives is assumed [12]. The assumptions of unbounded rationality are not met. FFTrees are an appropriate tool for handling the uncertain relationships between results obtained from multidisciplinary diagnostics and athletes' individual performance development. Given these circumstances, heuristics can maintain predictive accuracy while simplifying decision making by employing the less-is-more effect [13,14].

From a methodological perspective, heuristics have proven to be robust against overfitting [15,16]. In part, this could be due to their non-compensatory nature that provides "one-reason" decision making [15]. Each cue leads to a limited number of $n+1$ possible exits, with $n$ as the number of cues [17]. FFTrees are lexicographic, binary decision-making tools composed of search, stop, and decision rules [12,13]. The cues and their defined sequence form the search rule that needs to be identified. The identified sequence of cues represents the prioritization of variables for the decision according to their validity, thereby giving priority to the cue with the highest probability for a prediction [13]. The decision is terminated via a cue that allows a classification defined by the cut-off value through the stop rule, ignoring all subsequent lower-level cues [16]. FFTrees are therefore considered frugal [13]. The decision rule informs, in our case, which type of intervention is most important to consider [9]. FFTrees have the capability of facilitating prioritization among the various multidisciplinary areas of intervention.

As a tool for making decisions under uncertainty, FFTrees have been applied successfully within different domains, supporting multidimensional decisions such as medical diagnostics [18], the law [19,20], financial markets [21,22], traffic safety [23], and insurance [24]. An illustrative example is the medical decision as to whether a patient with symptoms possibly indicative of a heart attack should be sent home to a regular nursing bed for further monitoring or to a coronary care unit. In the first FFTree application, Green and Mehr [18] showed that anomalies in segment changes and a high value in the patient's electrocardiogram are sufficient to decide in favor of a coronary care unit. Work on such trees informed us to develop FFTrees in the context of elite sports while adhering to central principles formulated by the school of simple heuristics—namely, to create precise models with comparative and real-world testing [12,14] and to pursue the research objectives of heuristics outlined by Raab and Gigerenzer [9]. There is still a research gap in sports regarding how FFTrees can be applied to individual performance development and whether they can be validated through comparisons to alternatives in intervention selection.

Therefore, our study will deliver a proof of concept by developing FFTrees and evaluating their ability to determine individualized needs for interventions from multifaceted diagnostics in elite sports. Because different sports feature different specific characteristics and requirements, the decision for an intervention should be evaluated for each athlete and for each sport discipline. In this article, we consider differences between individual and team sports to be most fundamental, and we therefore validated sport-specific FFTrees for trampoline and volleyball to represent these two fundamental categories. In summary, the study overall aims to create sport-specific FFTrees as an intuitive decision tool to inform personalized training interventions in elite athletes.

More specifically, to achieve this aim, we showcase the reduction of complexity resulting from reducing multidisciplinary diagnostics to a few components in order to derive types of interventions facilitating the selection of variables used as cues for FFTrees. Following this data-driven reduction, we develop sport-specific FFTrees that allow decisions upon interventions for elite athletes at an individual level by assessing the derived multidisciplinary variables: the cues. In addition, we evaluate the quality of FFTrees not only with regard to their predictive accuracy but also by comparing them against other decision models.

## 2 Methods

### 2.1 Participants

This study included 466 German athletes (age $M = 18.93 \pm 4.19$ years, range: 13–39 years; 46% female) from different Olympic sports disciplines, including team and individual sports. Two sport-specific FFTrees were created on the subsamples of 27 trampoline gymnasts and 53 volleyball players for whom competition data were available. The analysis consisted of data from the nationwide in:prove project (funded by the German Federal Institute of Sport Science), a long-term collaborative research project running from December 2021 to August 2028. Data acquisition took place at junior or senior national-team training camps to which participating athletes had been assigned by their national-team coaches. These athletes were members of the top three national teams of the German elite sports system, containing junior squads (just started competing at international level), prospective squads (in preparation for next Olympics or chance for a medal), and Olympic squads (qualified for Olympics). As the nomination of squads in the German elite sports system is an annual procedure, the recruitment period is a continuous process. For our study, the recruitment phase started on 10/05/2022. All athletes who participated in the diagnostics until 14/12/2023 were included in the data analysis. Prior to the diagnostics, participants and, if necessary, their parents, provided written informed consent in accordance with the Declaration of Helsinki. The study protocol was approved by the Ethics Committee of the Faculty of Medicine of Justus-Liebig-University Giessen (ethical approval number: AZ 55/22; approval date: May 10, 2022).

### 2.2 Diagnostics

The analyses involved multidisciplinary diagnostics, including psychosocial, motor performance, and cognitive diagnostics. Furthermore, the included variables were trainable and contained a sufficient number of cases. The following sections describe the diagnostics included in the analyses.

**2.2.1 Psychosocial diagnostics.** In line with the theoretical model [7], psychosocial aspects were treated as environmental factors and assessed with psychosociological questionnaires (for further details, see Hilpisch et al. [25]). All questionnaires were used in the validated German versions. The Athlete Burnout Questionnaire (ABQ; [26] version Ziemainz et al., [27]) measured perceived burnout using a 5-point Likert scale with higher values representing higher perceived burnout in sports in the athlete. The Affect Balance Scale by Bradburn [28] measured the ratio between positive and negative emotions (i.e., hedonic balance) for subjective psychological well-being, with higher values representing more positive emotions. The Multidimensional Scale of Perceived Social Support (MSPSS; [29]) evaluated the perceived social support athletes received from family, friends, and significant others. A sport-specific cohesion questionnaire for individual and team sports (KIT-L, German: Kohäsionsfragebogen für Individual- und Teamsport – Leistungssport; [30]) measured group cohesion within a team or training group. Additionally, a specially developed questionnaire assessed general life satisfaction with the subscale non-sport satisfaction with one item in each of the areas of finances, private relationships, work, and leisure.

**2.2.2 Motor-performance diagnostics.** Motor performance was captured by a selection of sport-specific, performance-related measurements (see S1 File for the sport-specific selection). For mobility, the Y-Balance Test (YBT) for lower extremities and the Knee-to-Wall Test (KtW; [31]) were performed. Composite score of the right leg in the YBT, as well as the distance to the wall [cm] in the KtW, were evaluated.

Jump diagnostics consisted of the countermovement jump (CMJ; [32]) and drop jump (DJ) both assessed via OptoGait (Microgate, Bolzano, Italy). Jump height [cm] in CMJ and reactive strength index (RSI; Jump height/ground contact time) [m/s] in DJ were calculated. A 10 m sprint was conducted using the procedure described in Reichert et al. [33]. Sprint times [s] were captured using Microgate timing gates (Bolzano, Italy).

Grip strength was measured using a hand dynamometer (MicroFET®2, Hoggan Scientific LCC., Utah, United States) according to the procedure in Reichert et al. [34]. Absolute strength [N] values were captured, and relative strength (to body weight) was calculated [N/kg].

Lower-body tapping performance is a lower-body speed-related component in which athletes need to generate as many contacts as possible in an alternating manner on a contact mat (Voß, Doberschütz, Germany) over a 5-second duration, quantified by the maximum tapping frequency [Hz].

Further, dual task and motor inhibition diagnostics were conducted that incorporate both motor and cognitive performance aspects. Athletes performed a dual-task trial combining the tapping and a Stroop task [35] in accordance with the procedure described in Brinkbäumer et al. [36]. Motor dual-task costs [Hz] were subsequently calculated as the difference in tapping performance between the dual- and single-task conditions. Stop-signal reaction time to measure motor inhibition [ms] was quantified using the stop-signal paradigm by Verbruggen and Logan [37] and analyzed via the script developed by Verbruggen et al. [38] using the software MATLAB® (Version R2020a MATLAB 9.8).

**2.2.3 Cognitive diagnostics.** Cognitive diagnostics captured basic cognitive functions. While the trail-making task A (German equivalent: Zahlen-Verbindungs-Test, ZVT [39]) was used to measure information-processing speed, the d2-R [40] measured visual selective attention (i.e., concentration).

**2.2.4 Competition data.** To evaluate each athlete's performance, competition data from sport associations' databases were aggregated and averaged over 12 months. Each athlete received an individual performance value for the year 2023. A sport-specific variable was selected that was accurately descriptive of the sport discipline and could differentiate the individual athlete's performance in team sports. For the sport disciplines trampoline and volleyball, the following variables accommodated these requirements:

The total score in a trampoline competition is created based on five ratings referring to difficulty, execution, horizontal displacement, and flight time [41]. The competition data are gathered from international competitions listed in the Fédération Internationale de Gymnastique (FIG) database and the finals from the German Championships. The win–loss ratio (W-Lrat) in volleyball, derived from game reports, measures the ratio of gained and lost points from the actions of each athlete.

## 2.3 FFTrees

In preparation for the analysis, experts in the participating disciplines determined a selection of variables from diagnostics in in:prove that they deemed to be performance-related for their discipline. Additionally, the competition data had to be dichotomized into a logical variable to accommodate the binary decision of the FFTree assessing the need for intervention. In the first step, variables were analyzed by a principal component analysis (PCA) to identify cues for training the FFTrees in two sports disciplines. In the second step, two sport-specific FFTrees were trained and evaluated on trampoline and volleyball. Participants were 27 trampoline athletes (age $M = 18.87 \pm 4.20$ years, range: 14–30 years; 52% female) and 53 volleyball players (age $M = 20.81 \pm 5.86$ years, range: 14–36 years; 41% female). The need for intervention represented the outcome variable of the FFTrees, measured indirectly by the competition data of each athlete.

**2.3.1 Identification of cues.** To identify different types of potential interventions across sports among the multidisciplinary variables, we conducted a PCA with varimax rotation over the entire in:prove sample (see S2 File). This reduction ensured that the number of cues did not exceed the requirements of the FFTree algorithm (< 20 recommended) while maintaining variance in the data and allowing identification of groups of variables [42]. Variables with the highest loading within a component and the fewest missing values in a sports discipline were considered diagnostically representative for the type of intervention and therefore used as cues for creating FFTrees.

**2.3.2 Creation of FFTrees.** Because interventions need to capture sport-specific requirements, we constructed two sport-specific FFTrees as examples for both an individual and a team sport. From the sport disciplines tested in in:prove, trampoline and volleyball were chosen because of their large sample size and the wide diagnostic spectrum in their respective category. The subsamples for those athletes with competition data consisted of 27 trampoline gymnasts and 53 volleyball players. Inputs to the constructed FFTrees were cues determined by the PCA.

The FFTree itself was designed to predict whether an athlete is or is not in "need for intervention" by being trained on the input cues. As argued earlier, the interval-scaled competition variable was partitioned into a logical value (TRUE/

FALSE) based on distribution graphs that classified athletes into low or high performers to represent the "need for intervention." Low performers under the threshold mapped directly to a "need for intervention," whereas athletes above the threshold were defined as not in need for an intervention.

In addition, for trampoline, international qualification competitions for the Olympics found on the FIG database [43] were considered to determine the threshold. For both selected competitions—the 37th FIG Trampoline Gymnastics World Championships in Birmingham and the Qualifying OG for the FIG World Cup 2024—the 50th place in the qualification round was used as a reference value to guide the selection of the threshold. The volleyball threshold was aligned with the trampoline threshold for comparison reasons. To accommodate for differences in competition performance regarding the playing positions in team sports, partitioning was position-specific. The thresholds were established separately from considerations of sex and age, because competitions are only held and judged exclusively within gender and age groups.

The number of branches and how decisions are made in an FFTree can impact sensitivity and specificity [44]. Therefore, we used an algorithm that explores different combinations, because it facilitates weighing of sensitivity and specificity and allows inclusion of only decision-discriminating cues—called pruning—thereby optimizing the accuracy of the FFTrees [42]. To ensure efficiency and accuracy, the maximum number of cues was set to four [42].

Part of the data, in our case 60%, was used to train the FFTree algorithm. The FFTrees were tested on the remaining 40% of the data to estimate predictive performance. Partitioning slightly prioritized training data to ensure enough classifications with "no need for intervention" were included despite the small sample size and high threshold. Distribution was kept random and proportional to the classification. Athletes with missing values in a particular cue were ignored when analyzing that particular cue, but were included in all remaining cues. No imputation for missing data was performed, given that the other values remained unaffected and the sample size could be retained. Consequently, there was no risk of bias associated with the missing data. Accuracy of the FFTrees was assessed using correct classifications, sensitivity, and specificity, with results summarized in a confusion matrix. Additionally, to capture the frugality and efficiency of the decision-making process, the parameters percent cues ignored (pci) and mean cues used (mcu) were reported. To ensure reliable accuracy measurements, the FFTrees were evaluated through repeated random partitioning. For the model comparison, FFTree performance was compared to alternative classification algorithms.

### 2.4 Statistical analysis

We calculated z scores for each variable per sport discipline and sex, except for cognitive variables that were calculated only per sport discipline because their standard value already considers sex. Shapiro–Wilk tests were performed, and distributions were inspected for normality. Data processing and preparation were performed with Microsoft Excel for Mac (Version 16.86), and IBM SPSS Statistics 29 was used for the descriptive analysis. Data were represented by means and standard deviations.

For the first step, the PCA IBM SPSS Statistics 29 (©IBM Corp., Armonk, New York, USA) was used. In the second step, the FFTrees were created and analyzed using the package FFTrees 2.0.0 for the opensource Language R 4.2.2 [42,45].

## 3 Results

Six of the 17 z-transformed variables were normally distributed (Shapiro–Wilk test: $p > .05$). These were hedonic balance, relative grip strength, CMJ, YBT, tapping, and information-processing speed. A normal distribution was not a requirement for any of the statistics, and therefore did not lead to any limitations. Only the Bartlett test, which requires a normal distribution, was not considered when evaluating PCA.

### 3.1 Cue identification

Six components representing types of interventions were identified via explorative PCA with varimax rotation performed with correlations above $r = .30$ and a KMO test value of 0.63 (Table 1) for the sample of 466 athletes.

**Table 1. Rotated component matrix.**

| | Components | | | | | |
| | 1 | 2 | 3 | 4 | 5 | 6 |
|---|---|---|---|---|---|---|
| Burnout | | −0.75 | | | | |
| Social support | | 0.56 | | | | |
| Group cohesion | | 0.67 | | | | |
| Hedonic balance | | 0.76 | | | | |
| General life satisfaction | | 0.62 | | | | |
| Information-processing speed | | | 0.84 | | | |
| Visual selective attention | | | 0.81 | | | |
| CMJ | 0.78 | | | | | |
| DJ RSI | 0.74 | | | | | |
| 10 m sprint | −0.69 | | | | | |
| Tapping | 0.63 | | | | | |
| Motor cost | | | | | 0.82 | |
| Motor inhibition | | | | | | 0.93 |
| YBT | | | | 0.69 | | |
| KtW | | | | 0.86 | | |
| Rel. grip strength | 0.47 | | | | | |

Extraction method: Principal component analysis with varimax rotation. Component loadings exceeding values above .45 are presented. For all component loadings see S3 File; CMJ = countermovement jump, DJ = drop jump, KtW = Knee to Wall, rel. = relative, RSI = reactive strength index, YBT = Y-Balance Test.

The first component contained five variables (CMJ, DJ, sprint, tapping, and relative grip strength) belonging to the category of motor-performance diagnostics (focusing on power and speed diagnostics). Component two consisted of five variables from the psychosociological diagnostics (burnout, social support, group cohesion, hedonic balance, and general life satisfaction). Information-processing speed and visual selective attention, both cognitive diagnostics, loaded on the third component. Mobility diagnostics YBT and KtW were assigned to component four. The fifth component consisted of motor cost, which is indicative of dual task performance. Athletes with low motor cost can perform motor and visual cognitive tasks simultaneously without experiencing high interference between the tasks that produce motor task decrements. Motor inhibition loaded on the sixth component, representing the ability of athletes to interrupt and stop a planned action after a visual stimulus indicating to stop the movement.

Based on the components, resulting cues for the FFTrees representing types of intervention selected on component loading were CMJ, hedonic balance, information-processing speed, YBT, motor cost, and motor inhibition for volleyball. The same selection was applied to trampoline, except that visual selective attention replaced information-processing speed, and CMJ was exchanged for relative grip strength. The rationale behind this was the consideration of variables with the highest loading within a component as well as the fewest missing values.

### 3.2 Fast-and-frugal trees

Analysis of the competition variable—the total score ($M = 42.27 \pm 10.66$ points) in trampoline and the W-Lrat ($M = 2.19 \pm 3.16$) in volleyball—resulted in the following thresholds.

For comparison with international qualifying competitions, female athletes reached 49.40 and 50.86 points, and male athletes 54.71 and 53.96 points. These results correspond to the 75th percentile in the in:prove population (total score$_{female}$ = 48.43; total score$_{male}$ = 51.04).

 

Derived from the threshold level in trampoline, the 75th percentile was used for volleyball and defined for each playing position individually, leading to position-specific thresholds for W-Lrat. From 53 volleyball players, 23 played the position of outside hitter with the corresponding threshold of 5.38, 16 were middle blockers at 2.95, 10 were setters at 1.18, and 4 were opposite hitters at 6.00.

Based on the cue identification, two FFTrees were created for the subsamples of trampoline and volleyball. The FFTree with the highest predictive performance is presented for trampoline in Fig 2 and for volleyball in Fig 3.

In the FFTree for the individual sport trampoline ($n = 27$) with six cases of "no need for intervention," the decision was made by four cues (relative grip strength, motor cost, motor inhibition, visual selective attention). The FFTree achieved a sensitivity of 92% and specificity of 100% with a balanced accuracy of 96% for the training data (S4 File). Predictive

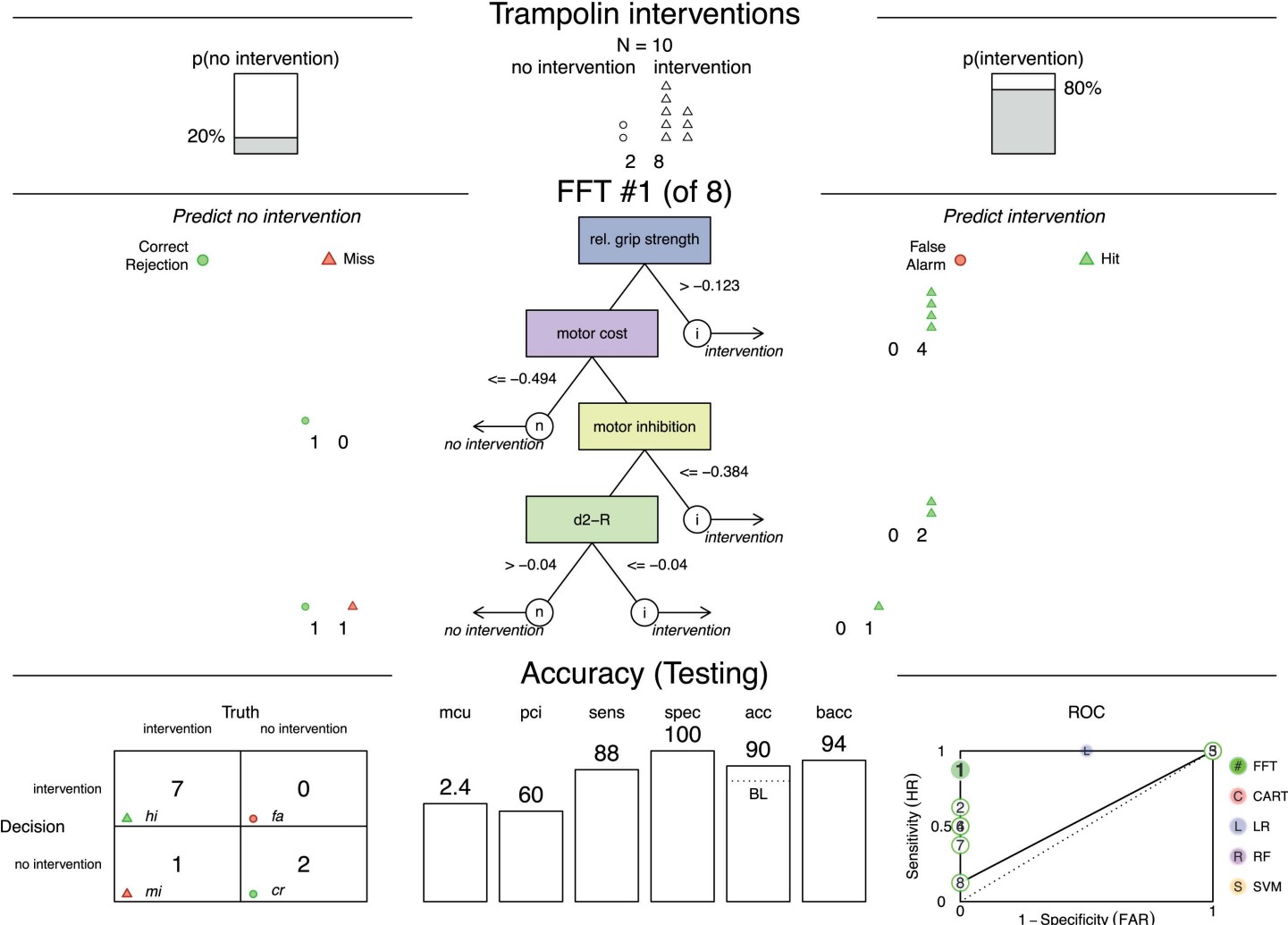

**Fig 2. Fast-and-frugal tree for trampoline intervention including predictive performance.** Fast-and-frugal tree (FFTree) to determine an intervention for trampoline gymnasts based on four cues including classification and predictive accuracy of test data from 40% of the subsample calculated with the ifan algorithm and the z scores; acc = accuracy, bacc = balanced accuracy, d2-R = d2-Test revised version (visual selective attention), mcu = mean cues used, pci = percentage cues ignored, ROC = receiver operating characteristic (presenting the performance of all FFTrees from the "fan" based on the bacc with false alarm rate [FAR] on the x and sensitivity on the y axis), sens = sensitivity, spec = specificity.

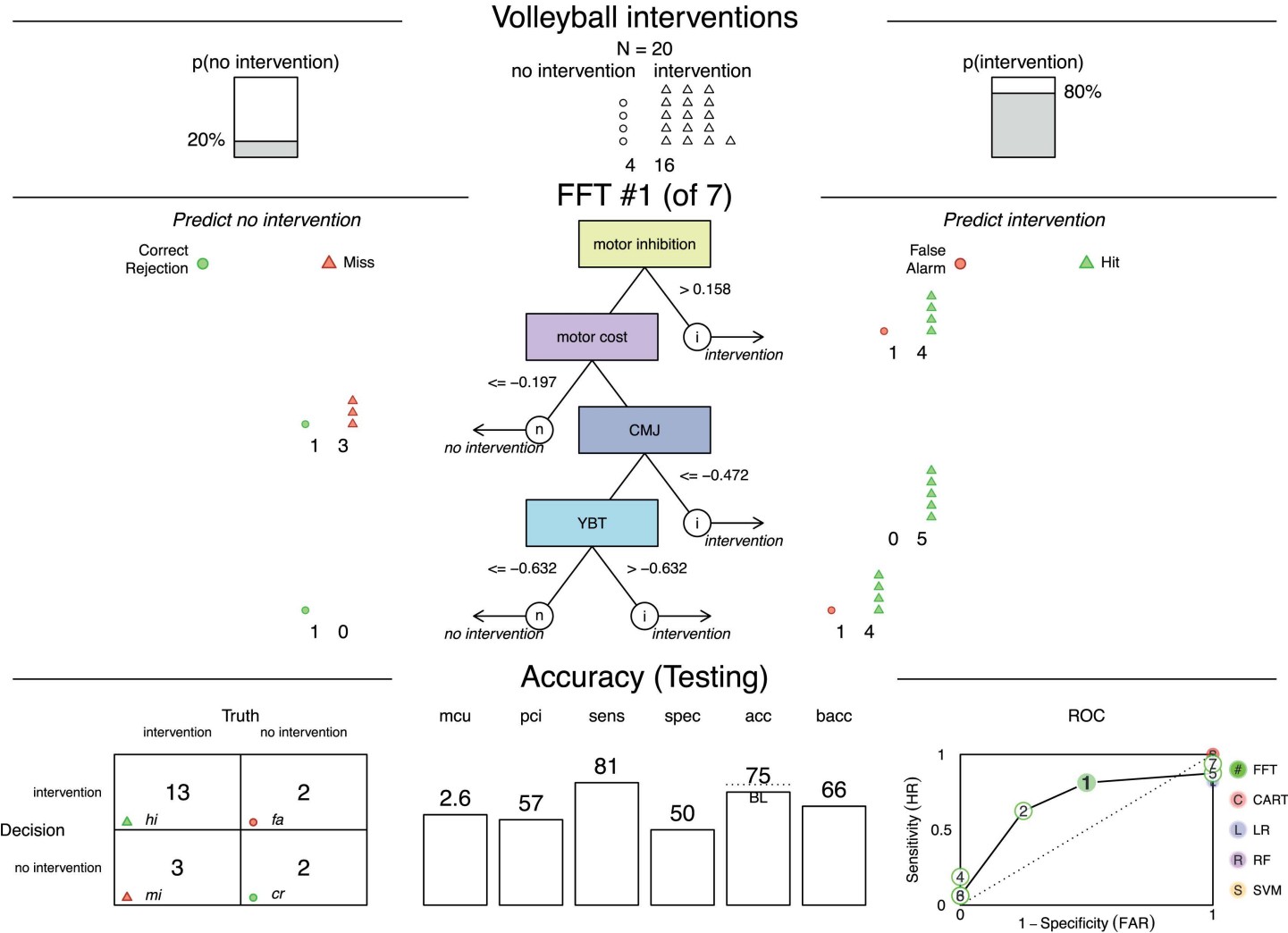

**Fig 3. Fast-and-frugal tree for volleyball intervention including the predictive performance.** Fast-and-frugal tree (FFTree) to determine an intervention for volleyball players based on four cues including classification and predictive accuracy of test data from 40% of the subsample calculated with the ifan algorithm and the *z* scores; acc = accuracy, bacc = balanced accuracy, CMJ = Countermovement Jump, mcu = mean cues used, pci = percentage cues ignored, ROC = receiver operating characteristic (presenting the performance of all FFTrees from the "fan" based on the bacc with false alarm rate [FAR] on the x and sensitivity on the y axis), sens = sensitivity, spec = specificity, YBT = Y-Balance Test.

performance is presented in Fig 2. On average, 2.4 cues were necessary for decision making; the subsequent 60% of cues were ignored. The decision-making process can be interpreted as follows, illustrated with the first cue: relative grip strength. The threshold suggests that once athletes' relative grip strength *z* values were above −0.12, they can be classified as needing intervention. All following cues are ignored. Scoring below this threshold will lead to the assessment of the motor costs. An individual motor cost *z* value less than or equal to −0.49 classifies athletes as not being in need for an intervention. Values above would indicate deficiencies in dual task performance. However, a definite decision will only be made in subsequent cues such as motor inhibition. The first cue, relative grip strength, was able to classify 40% of cases correctly as in "need for intervention".

The FFTree for volleyball as a team sport ($n = 53$), with 12 athletes with "no need for intervention," consisted of four cues (motor inhibition, motor cost, CMJ, YBT). The FFTree resulted in 48% sensitivity, 100% specificity, and a balanced accuracy of 74% on training data (S4 File). The decision process classified the athletes after 2.06 cues on average and ignored 57% of the information. Illustrated on the first cue, for a volleyball player with motor inhibition, this implies that above a $z$ value of 0.16, a need for intervention is identified. At lower values, motor costs are considered in the decision process. Athletes with motor costs less or equal to a $z$ value of −0.20 are not in need for an intervention. All athletes having values above −0.20 could show a decrement in dual task performance and subsequent cues must be considered. The threshold for CMJ suggests that athletes with a $z$ value below −0.47 are classified as needing intervention. Similarly, the threshold $z$ value of −0.63 for the YBT cue classifies athletes as requiring mobility interventions.

In the trampoline FFTree, only one cue contained a missing value from one athlete in motor cost. The athlete was therefore excluded for the analysis of the motor cost cue. In the volleyball FFTree, a quota of 5.6% missing values could be found in CMJ, hedonic balance, motor cost, motor inhibition, and information-processing speed. In the analysis of the respective cues, the missing values were omitted from the analysis.

Examination of the 20 randomly generated FFTrees (S5 File) consisted of different cues in various cue order and showed signs of pruning. For trampoline, in order of decreasing frequency, relative grip strength, motor inhibition, motor cost, hedonic balance, visual selective attention, and YBT were used for the decision. In volleyball, CMJ with 85% was the most used cue followed by motor cost, motor inhibition, information-processing speed, and YBT. In Fig 4 differences

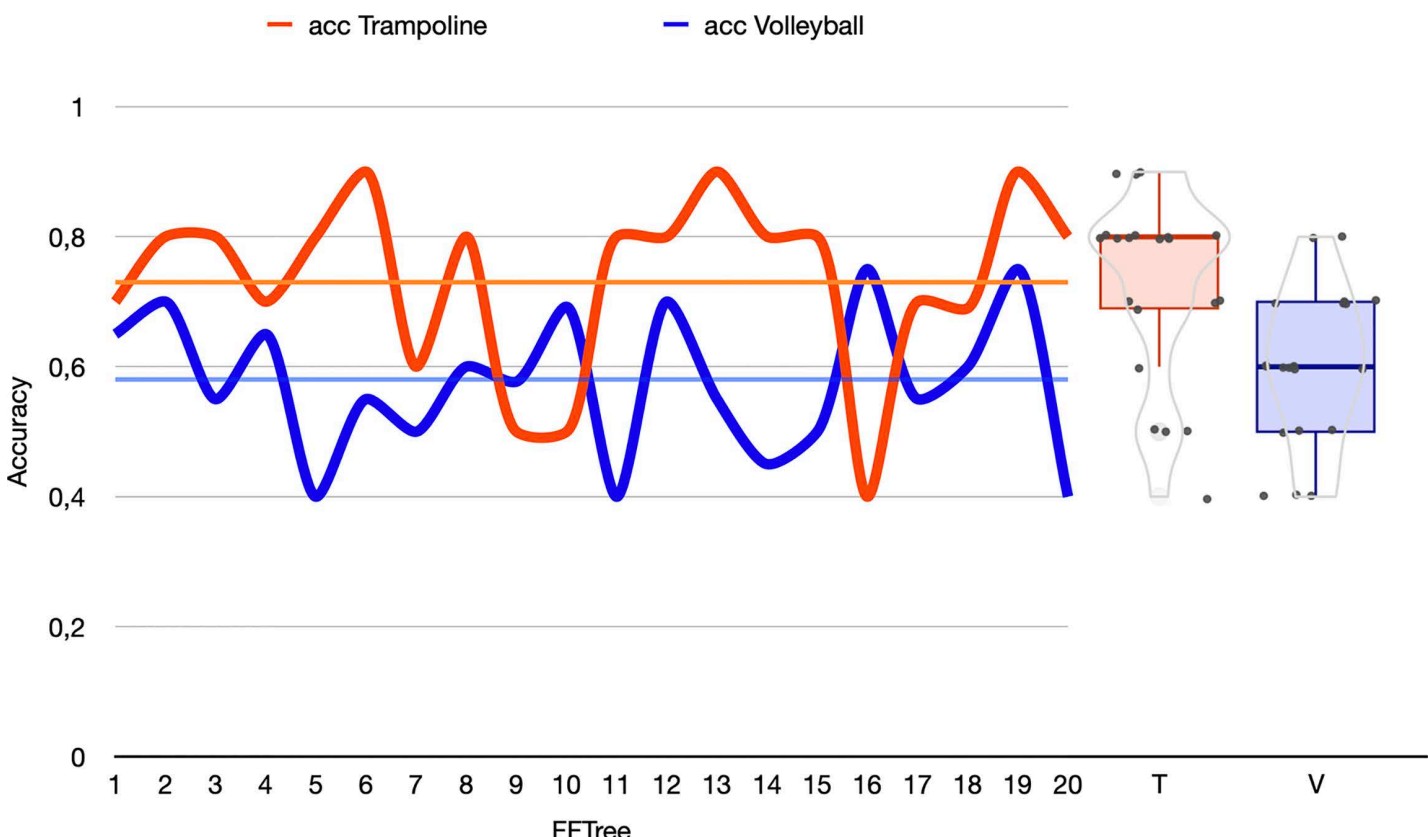

**Fig 4. Predictive performance of the FFTrees.** Predictive accuracy (acc) of FFTrees based on 20 randomly generated test data sets with 40% of the subsamples in trampoline (red) and volleyball (blue) and the box and violin plots. Mean of predictive values from both sport disciplines visualized by a reference line; acc = predictive accuracy, FFTree = fast-and-frugal tree, T = trampoline, V = volleyball.

between the predictive performance of the 20 random FFTrees of both sports disciplines can be seen with a variation between sensitivity (trampoline: $M = 0.78 \pm 0.14$; volleyball: $M = 0.64 \pm 0.17$) and specificity (trampoline: $M = 0.53 \pm 0.44$; volleyball: $M = 0.34 \pm 0.31$). FFTrees in trampoline achieved higher performance overall than volleyball for training data and testing data as can be seen in Fig 4.

FFTrees achieved higher predictive performance than alternative classification algorithms as visible in the ROC curve (seen in the lower right corner in Figs 2 and 3). The ROC curve gives a visual presentation of the sensitivity and specificity with superior performance located in the top left corner of the graph. FFTrees exceed alternative classifications in terms of sensitivity and specificity, indicating that decisions in both directions either for or against a "need for intervention" classified more correct predictions.

## 4 Discussion

The aim of this study was to build and evaluate sport-specific FFTrees informing individual intervention decisions. To identify the most promising types of intervention, heuristic FFTrees can be a helpful tool. One FFTree was created for trampoline (individual sport) and one for volleyball (team sport) based on diagnostics of elite athletes and current competition performance. This ensures real-world testing in accordance with the fifth principle of Raab [14] defined as "test heuristics in the real world." The creation of FFTrees incorporates biopsychosocial determinants, reflected in the selection of multidisciplinary cues [7]. FFTrees bridge the multidisciplinary understanding of expertise with individualized intervention selection. Due to the many diagnostic variables but limited number of elite athletes, small sample sizes are a critical characteristic. FFTrees are suitable tools in elite sports as we need to consider time and training constraints that require us to make decision on which athletes to prioritize for specific interventions. FFTrees rely on the concept of bounded rationality. This is based on Herbert Simon's [46] view addressing the limitations of our mind and the structure of the environment in which decisions unfold. Such bounded rationality perspective is not specific to FFTrees as evident for instance in Brunswik's [47] work or ecological approaches [48]. A difference between ecological approaches and heuristics, however, is that while ecological approaches consider person and environment as one system, heuristic programs consider them interacting components (see Raab & Araujo [49] for a discussion of differences in sports). FFTrees are able to identify cues and define cut-off values with a specific cue order in order to classify athletes successfully and to decide upon their individual type of intervention.

The results of our study suggest that FFTrees can interpret multidisciplinary diagnostics and lead to transparent decisions on interventions, supporting our proof of concept. The non-compensatory design of FFTrees makes it possible to identify a decision-relevant type of intervention with the most promising adaptation potential. As seen in Fig 3, athletes with lower performance in CMJ are selected to improve CMJ performance or generalized strength parameters. However, decisions for no intervention (e.g., motor cost) do not provide clear identification of adaptation potential. These decisions require the assessment of subsequent cues such as CMJ or YBT. Despite this, decisions in both directions are preferred to decisions for "intervention" only, because this maintains specificity and allows a differentiation of needs for intervention [10,42].

For the proof of concept, the quality of the FFTrees classifications was evaluated in terms of predictive accuracy and plausibility. Accuracy of the decision reached by FFTrees was evaluated based on three parameters: their predictive value, frugality, and speed [42]. Dataset partitioning makes it possible to assess testing data in order to detect overfitting. By evaluating how well FFTrees predict data, we follow the fourth principle of Raab [14] defined as "examine how well models of heuristics predict new data." The trampoline FFTree correctly classifies a remarkable 90% of needs for intervention, whereas the volleyball FFTree achieves 75% accuracy, suggesting that both are capable of accurately determining the need for interventions. Compared to alternative classification algorithms such as logistic regression, heuristic approaches have been demonstrated to achieve higher accuracy in predicting data in uncertain domains with small

sample sizes [18]. Our findings align with this sentiment in that FFTrees outperformed alternative classification methods. Although the created FFTrees show slightly lower accuracy in testing compared to training data (S5 File), this decline is to be expected. In contrast, overfitting is evident in the RF algorithm that attains optimal values in training data but cannot maintain this accuracy in testing data. This finding is consistent with prior research indicating that the RF algorithm is susceptible to overfitting, particularly when the training set is small [50]. Deeper models, such as artificial neural networks, require even more data. Empirical guidelines recommend a minimum of 10–20 observations per weight as necessary to ensure reliable generalization [51]. Additional analyses (i.e., ROC curves) demonstrates that in comparative testing in accordance with the second principle of Raab [14] defined as "test heuristics comparatively," FFTrees exhibit a lower risk for overfitting and enhanced generalization.

Due to the different demands of team and individual sports, we used trampoline and volleyball as a showcase to test FFTree performance. FFTrees reveal higher predictive accuracy for trampoline (73 ± 14%, max. 90%) than for volleyball (58 ± 11%, max. 75%.) despite the smaller sample size in the former. The disparity in FFTree performance between individual and team sports indicates that the FFTree for trampoline can more effectively capture the structure of the performance determinants. In volleyball, however, the FFTree is not quite capable of correctly classifying all volleyball players using the same set of decision rules. This discrepancy may be due to the more complex and uncertain structure of team sports such as many different games systems (i.e., tactics), and player positions that differ in their requirements and the respective athlete types [52–54]. Unlike individual sports, where every athlete is confronted with the same athletic requirements and performance goals, team sports are a collection of different player positions with different tasks specific for their position. It is likely that a single FFTree may not adequately represent the heterogeneous nature of volleyball and other team sports, but that position-specific FFTrees might provide a more accurate solution when deciding on the individually most promising intervention. This approach would require larger sample sizes. FFTrees demonstrate the less-is-more effect by including only cues that contribute adequately to the decision and allowing pruning without limitations on accuracy (S5 File). Frugality and speed are evident with both FFTrees allowing decisions with an average of 60% ignored information.

The *plausibility* of FFTrees was evaluated by comparing them to known findings and the prioritization by coaches. Performance in diagnostic test, grip strength, and CMJ can differentiate significantly between successful and unsuccessful volleyball players [55]. Both sports disciplines value cognitive skills [56,57], reflected in FFTrees through low motor cost classifying no need for intervention. Athletes with high competition performance can perform motor and visual cognitive tasks simultaneously without experiencing high interference in the motor task. In trampoline, high visual selective attention is related to better competition scores. That the FFTrees are plausible can be seen from a good reflection of relevant cues mirroring the performance demands of the respective sports. Expanding the comparison to the multidisciplinary priority list created by coaches of the sport (S6 File) shows that both disciplines assign high priority to the motor diagnostics represented in the FFTrees (Fig 2). PCA embodies this prioritization with three of the six defined components representing motor-performance determinants (speed and strength component, mobility component, motor inhibition). Both FFTrees include two out of three components. The second-highest priority in volleyball is cognitive determinants. Both FFTrees contain motor inhibition and motor cost as cues. Psychosociological determinants have a higher priority in trampoline than volleyball. Though absent in selected FFTrees, they appear in 55% of the 20 randomly generated FFTrees in volleyball, and in half of those in trampoline. Reviewing the frequency of cues used in the random generated FFTrees shows similar tendencies to the prioritization, which encourages the assumption that larger samples could lead to more "stable" FFTrees. The alignment between the profile of requirements and the priority list of coaches to the FFTrees reinforces the plausibility of the FFTrees.

In practice, a coach reviewing the FFTree can gather information regarding which areas for intervention are most important for potential performance improvements. This can be seen in the order of the cues within the FFTree. The higher the cue, the more important it is. For instance, strength-based interventions merit high priority if the FFTree identifies relative grip strength

or CMJ as a first-level cue. Furthermore, the cut-off values offer norm values appropriate for elite athletes. The primary benefit coaches derive from FFTrees is the ability to directly assess and categorize diagnostic information from athletes, thereby informing decisions regarding training regimes. Consider the volleyball player A, who exhibits high motor inhibition and low motor costs. However, his CMJ performance is below the threshold. In this scenario, the FFTree suggests the implementation of strength interventions within the athlete's training regime to enhance performance. The performance of athlete A in the YBT would not affect the decision and can be ignored. The intervention with the highest priority for the athlete is identified. The FFTree can be used again to find additional intervention needs if capacities for further interventions are available.

The multiple randomly generated FFTrees show *variability in characteristics*, with cues and cue order depending on data partitioning. In the following, this phenomenon is referred to as "unstable" FFTrees. Consequently, the FFTrees created for trampoline and volleyball represent possible solutions rather than definite ones. Therefore, a sound methodological concept for feature selection in the initial design stage of FFTrees needs to be implemented. A comprehensive understanding of the requirements inherit to the sport can inform which variables should be incorporated into the FFTree creation process. The use of a PCA can provide statistical information on which areas of intervention exist and the diagnostic that best represent them. Additionally, we recommend cross-validating the created FFTrees with established research findings to ensure their plausibility. Notwithstanding the assumption and the presented results that FFTrees achieve high accuracy in small sample sizes, it can be assumed that a larger, more homogeneous sample would lead to stable FFTrees with a distinct cue order—a tendency seen in other datasets [42]. Distinct cue orders could give an indication on prioritization of the multidisciplinary diagnostics for each sports discipline. Previously generated FFTrees [24,42,58] used considerably larger samples than the present study. One potential solution to the inevitable small sample sizes in elite sports might be the aggregation of data from multiple years, thereby creating a more substantial database. This would provide an increasing cohort for generating and refining FFTrees.

Construction of FFTrees incorporates the *search rule*, considerations on sensitivity and specificity, and a dichotomization of competition performance. The search rule was implemented methodologically through the PCA that successfully identifies the various disciplines and differentiates motor skills through components. PCA attributes multidisciplinary variables to components eligible for FFTree cues while adhering to heuristic principles. Variables with the highest component loadings and fewest missing values are selected as cues, maintaining transparent cues and intuitive cut-off values. To avoid distortions, missing values were not replaced with mean values. Managing missing values is an essential challenge in elite sports. Injuries and other health-related concerns can limit participation in diagnostic procedures. Using $z$ scores as cut-off values enables the generation of a single FFTree per sport discipline that is sensitive to sex differences.

Consideration of *sensitivity and specificity* is important when managing potential risk. Sensitivity was prioritized over specificity due to the lower harm of unnecessary interventions compared to omitted adaptation opportunities. Nevertheless, athletes' time resources are limited, and, thus, specificity remains relevant. With sufficient resources, FFTrees can be used iteratively to identify multiple intervention needs. The pronounced sensitivity (> 80%) confirms that both FFTrees generate little omitted adaptation potential.

*Dichotomization* of competition performance is essential for categorizing "need for intervention." Optimal performance level, defined as not necessitating intervention, includes athletes with various squad affiliations, ruling out potential mediators such as age. The 75th percentile was selected as a conservative threshold to minimize undetected adaptation potential in accordance with sensitivity and specificity considerations. Derivation of thresholds based on international competition results ensures international relevance and addresses potential selection bias. After all, in elite sports, not only individual improvements but competitiveness on an international scale are of great importance, as evidenced by the Olympics.

The findings of this study have to be seen in light of certain limitations. For team sports, FFTrees may require position-specific trees to adequately represent the heterogeneous characteristics of different positions in volleyball and other team sports. Given the current sample size, implementing this approach is not yet feasible. Additionally, FFTrees exhibit instability across random partitions. A possible solution could be increasing the sample size and

expanding the data on which FFTrees are created. Numerous junior squad volleyball players were excluded due to a lack of individual performance records at their competition level. Generalization to these squad levels as well as to non-German athletes should be realized with caution. Hence, a comprehensive individualization approach is required that not only encompasses the diagnostics and interventions but also includes methods for performance development and competition diagnostics that differentiate individual contributions in team settings.

Efficient performance diagnostics requires the art of focusing on the essentials and ignoring the rest. Boullosa [59] recommended machine learning and artificial intelligence to support decision-makers. FFTrees align with this sentiment by translating the amount of data by identifying performance-related diagnostics for individualized training approaches. The intuitive, transparent design increases comprehension and acceptance among decision-makers and athletes [18,60]. Athletes can benefit from individualized interventions through achieving their performance potential, increased intrinsic motivation, and improved long-term development [5,61]. Individualized interventions are associated with injury prevention addressing individual deficits and risks, while potentially reducing overtraining-related injuries through improved training efficiency [5,59].

Previous research has demonstrated the versatility of applications for FFTrees [18–24]. Additionally, Raab and Gigerenzer [9] have proposed FFTrees as a helpful approach to performance science. Consistent with this, our study has demonstrated the efficacy of FFTrees as a decision-making tool in elite sports. In order to expand upon our research and recent findings, we advocate the implementation of sport-specific and position-specific FFTrees in additional sports disciplines. The establishment of a transparent protocol for the selection of interventions tailored to the needs of athletes in various sports would be promoted by this. Simultaneously, it would provide insights into the requirement profiles and prioritization of multidisciplinary diagnostics for the respective sport.

Future research should focus on developing improved FFTrees that determine types of intervention by considering individual prerequisites (e.g., anthropometric data, response patterns, and genetic data), expanding them to more sports disciplines, and evaluating intervention effects on competition performance longitudinally.

## 5 Conclusions

The current findings findings suggest that FFTrees serve as a practical and efficient decision-making tool in sports science, offering an approach to unlock athletes' true adaptation potential. This innovative method not only supports coaches and sports scientists in identifying athletes' adaptation potential based on an individual holistic approach but also expands the landscapes of performance evaluation and improvement. The alignment between FFTrees with the corresponding priority list from coaches and sport-specific requirements supports and validates their plausibility. FFTrees enable an efficient interpretation of extensive diagnostic data, making individualization more manageable and efficient. In a demonstration of their capabilities, FFTrees outperform alternative classification algorithms, cementing their position in sports performance analysis.

The development of FFTrees illustrates their suitability as precise decision models for sports-scientific data, particularly in elite sports characterized by small sample sizes, missing values, and the inherently dynamic, uncertain nature of athletic environments.

FFTrees achieve higher accuracy in individual sports compared to team sports, suggesting that position-specific FFTrees could significantly improve decision-making in the latter. Expanded research with larger sample sizes could produce more stable FFTrees and lead to position-appropriate intervention decision.

## Supporting information

**S1 File. Descriptives for all variables in whole sample and subsamples.** No values indicate diagnostics were not applied in this discipline, Exclusion criteria include high missing values (°), "first step" interventions (*); ᵃ = only female athletes, AR/IR = external/internal rotation, CMJ = countermovement jump, d2-R = d2-Test revised version (visual selective attention), DJ = drop jump, D/ND = dominant/ nondominant side, HAST = Handball Agility-Specific Test, KtW = Knee

to Wall, KV = strength ratios, rel. = relative, RJ = Repeated Jumps performance decrement, RSI = reactive strength index, Schd = shoulder diagnostics, YBT = Y-Balance Test, ZVT = Zahlenverbindungstest (information-processing speed).
(DOCX)

**S2 File. Principal component analysis.**
(DOCX)

**S3 File. Rotated component matrix.** Extraction method: Principal component analysis with varimax rotation; CMJ = countermovement jump, DJ = drop jump, KtW = Knee to Wall, rel. = relative, RSI = reactive strength index, YBT = Y-Balance Test.
(DOCX)

**S4 File. FFTree with training dataset including equation for R.** FFTree for trampoline generated for the training dataset with 60% of the subsample via the formula: FFTrees(formula = WKOlymp ~ ZL6_HedBalance + ZKL_SW +. ZGriffk_Re_rel + ZYBAL_Composite_Re + ZMotCost_TapStr + ZInhi_SSRT_Hand, data = HeuristicTreeTrampolin, train.p = 0.6) and FFTree for volleyball generated with the training dataset with 60% of the subsample via the formula: FFTrees(formula = WKOlymp ~ ZL6_HedBalance + ZZVT_SW + ZCMJ_bb + ZYBAL_Composite_Re + ZMotCost_TapStr + ZInhi_SSRT_Hand, data = HeuristicTreeVolleyball_Lib, train.p = 0.6); acc = Accuracy, bacc = balanced accuracy, d2-R = d2-Test revised version (visual selective attention), CMJ = countermovement jump, mcu = mean cues used, pci = percentage cues ignored, ROC = receiver operating characteristic (presenting the performance of all FFTrees from the "fan" based on the bacc with false alarm rate [FAR] on the x and sensitivity on the y axis), sens = sensitivity, spec = specificity, YBT = Y-Balance Test.
(PDF)

**S5 File. Randomly generated FFTrees.** Twenty random generated FFTrees for trampoline as well as for volleyball and their performance are represented for the training and test data as well as mean (M) and standard deviation (SD), used cues are listed in descending order; acc = accuracy, bacc = balanced accuracy, CMJ = countermovement jump, d2-R = d2-Test revised version (visual selective attention), HedBalance = hedonic Balance, Inhibition = motor inhibition, mcu = mean cues used, MotCost = motor cost, pci = percent cues ignored, Sens = sensitivity, Spec = specificity, YBT = Y-Balance Test, ZVT = Zahlenverbindungstest (information-processing speed).
(DOCX)

**S6 File. Priority lists made by coaches.** Descending prioritization for the multidisciplinary diagnostics by the coaches from DTB and DVV. The menstrual cycle is only considered for female athletes; DTB = Deutscher Turner-Bund, DVV = Deutscher Volleyball-Verband.
(DOCX)

## Acknowledgments

We cordially thank Jonathan Harrow for native speaker advice and all our athletes for participating in this study.

## Author contributions

**Conceptualization:** Lena Siebert, Karen Zentgraf.

**Data curation:** Lena Siebert, Lukas Reichert, Lisa Musculus, Laura Will.

**Formal analysis:** Lena Siebert.

**Funding acquisition:** Markus Raab, Karen Zentgraf.

**Investigation:** Lena Siebert, Lukas Reichert, Lisa Musculus, Laura Will, Karen Zentgraf.

**Methodology:** Lena Siebert, Lukas Reichert.

**Project administration:** Lukas Reichert, Markus Raab, Karen Zentgraf.

**Resources:** Markus Raab, Karen Zentgraf.

**Software:** Lena Siebert, Ahmed Al-Ghezi.

**Supervision:** Karen Zentgraf.

**Validation:** Lukas Reichert, Karen Zentgraf.

**Visualization:** Lena Siebert.

**Writing – original draft:** Lena Siebert.

**Writing – review & editing:** Lena Siebert, Lukas Reichert, Lisa Musculus, Laura Will, Ahmed Al-Ghezi, Markus Raab, Karen Zentgraf.

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
