## [Decision Letter · Decision Letter 0]

6 Jun 2025

PONE-D-25-17971Fast-and-frugal trees: A decision tool for elite sportsPLOS ONE

Dear Dr. Siebert,

Thank you for submitting your manuscript to PLOS ONE. After careful consideration, we feel that it has merit but does not fully meet PLOS ONE’s publication criteria as it currently stands. Therefore, we invite you to submit a revised version of the manuscript that addresses the points raised during the review process.

We look forward to receiving your revised manuscript.

Kind regards,

Mário Espada, PhD

Academic Editor

PLOS ONE

Journal Requirements:

3. Thank you for stating the following financial disclosure: [This work was supported by the Bundesinstitut für Sportwissenschaft (German Federal Institute for Sport Sciences) in 2021–2025 under Grant No. 081901/21-25 (English title: Individual performance development in elite sports by holistic and transdisciplinary process optimization). The data analyzed for the purposes of this study were part of a multidisciplinary large-scale data set that included multiple points of measurement and a cross-sectional and longitudinal perspective. The subset of data included in this study covered the cross-sectional data collected in the period 01/2022–12/2023.].  

Please state what role the funders took in the study.  If the funders had no role, please state: """"The funders had no role in study design, data collection and analysis, decision to publish, or preparation of the manuscript."""" 

4. In the online submission form, you indicated that [The raw data supporting the conclusions of this article will be made available by the authors upon request, without undue reservation.]. 

5. We are unable to open your Supporting Information file S4.1 trampoline.eps and S4.2 volleyball.eps. Please kindly revise as necessary and re-upload.

Additional Editor Comments:

Dear Authors,

Please revise the manuscript considering the reviewers´ suggestions.

Thank you.

Best regards.

Reviewers' comments:

Reviewer's Responses to Questions

**Comments to the Author**

1. Is the manuscript technically sound, and do the data support the conclusions?

Reviewer #1: Yes

Reviewer #2: Yes

Reviewer #3: Yes

2. Has the statistical analysis been performed appropriately and rigorously? 

Reviewer #1: Yes

Reviewer #2: Yes

Reviewer #3: Yes

3. Have the authors made all data underlying the findings in their manuscript fully available?

Reviewer #1: Yes

Reviewer #2: Yes

Reviewer #3: Yes

4. Is the manuscript presented in an intelligible fashion and written in standard English?

Reviewer #1: Yes

Reviewer #2: Yes

Reviewer #3: Yes

5. Review Comments to the Author

Reviewer #1: The use of Fast-and-Frugal Trees (FFTs) in sports performance forecasting represents an interesting methodological advance that addresses the need for rapid, transparent, and efficient decision-making in contexts where time and clarity are critical, such as sports. Unlike complex statistical models or deep learning algorithms, FFTs offer a simplified decision structure that prioritizes speed and straightforward interpretation, which is especially valuable for coaches.

One of the study's key strengths is its ability to balance accuracy and usability. FFTs are easily understandable and applicable in real-world settings without requiring advanced software, which favors their implementation in sports clubs, youth national teams, and development programs. Furthermore, their explicit, hierarchical design facilitates communication between analysts and coaching staff.

However, it is important to recognize certain limitations inherent to the model. While FFTs perform well in binary decision settings or when speed is prioritized over complexity, they may have lower predictive capacity than more robust models (such as Random Forests or neural networks) in contexts with nonlinear data or many interactions between variables. Furthermore, their performance can be sensitive to the initial feature selection, which requires methodological care in the design stage.

This study constitutes a valuable contribution to the field of sports performance by demonstrating that it is possible to build agile and accessible predictive models without sacrificing their practical utility. The use of FFTs invites us to rethink how informed decisions are made in everyday sports contexts, highlighting that heuristic simplicity can be a virtue in complex environments, provided it is supported by rigorous empirical evidence.

Reviewer #2: Title of Manuscript: Fast-and-frugal trees: A decision tool for elite sports

Manuscript ID: PONE-D-25-17971

Reviewer’s report

First, I would like to warmly acknowledge the considerable effort behind this manuscript. The multidisciplinary approach, the ambition to simplify complex decision-making through FFTrees, and the relevance to elite sports settings are evident throughout the work. My comments aim to support the authors in refining the manuscript to meet the high scientific and editorial standards required by PLOS ONE. I have structured the feedback section by section, with explicit suggestions that I hope will be helpful in moving the paper forward.

1. Abstract

Suggestion: The structure could be improved for clarity. Consider explicitly segmenting the abstract into: Objective, Methods, Results, Conclusion.

For example:

Line 6–7: You may consider beginning with a clear objective such as: “The objective of this study was to examine the applicability of fast-and-frugal trees (FFTrees) to inform training interventions in elite athletes.”

Clarity of FFTrees: As this might be unfamiliar to some readers, consider briefly defining what FFTrees are in one sentence near the start.

2. Introduction

Objective not clearly stated:

Lines 101-104: The objective appears but is somewhat embedded. You may consider adding a clear and concise sentence like:

“The aim of this study is to develop and evaluate the predictive performance of sport-specific FFTrees to inform personalised training interventions in elite athletes.”

Theory clarification:

Lines 31-40: You refer to theoretical models but do not fully explain how these support the rationale for FFTrees. Consider integrating the constructivist or biopsychosocial lens more explicitly to frame why FFTrees are an appropriate tool for this context.

Suggestion for structure: You might consider adding subheadings or clearly defined paragraphs separating:

Context and problem (elite sports complexity)

Existing gaps in intervention selection

Introduction to FFTrees as a methodological response

3. Methods

Cue Selection Explanation:

Lines 210-225: The PCA is well described but consider clarifying why each variable was selected as a cue. For example, in line 299-301, you could briefly mention the rationale for replacing information-processing speed with visual selective attention in trampoline.

Clarify Dichotomisation Criteria:

Lines 234-245: Please make the thresholding process more transparent. For instance:

How was the “75th percentile” decided?

Were sex and age considered in the dichotomisation?

Reproducibility:

Lines 272-273: You may consider adding the R script or relevant code as supplementary material or sharing a GitHub link, in line with open science practices.

4. Results

Table 3.1 (line 293): It would be helpful to add a brief sentence interpreting the practical relevance of each component, what does “motor cost” imply, for instance?

Figures 3.1 and 3.2:

The decision rules could be better explained in-text. For example, in line 327-328, you might add:

“This threshold suggests that athletes with grip strength z-scores > -0.12 are classified as needing intervention, reflecting possible strength deficiencies.”

Missing data handling:

Line 256-258: Please state how many cases were excluded due to missing data, or clarify if imputation was considered.

5. Discussion

Theoretical anchoring:

Lines 382-390: While you reference the multidisciplinary and heuristic value of FFTrees, it would strengthen the paper to briefly connect to models such as constructivist decision-making or ecological dynamics, especially since you touch on bounded rationality. This would deepen the theoretical contribution.

Practical implications:

Lines 445-455: Consider adding an example of how a coach could use a FFTree in a training context. For example:

“A coach reviewing FFTree output could prioritise strength-based interventions if the tree identifies grip strength or CMJ as first-level cues.”

Limitations:

While limitations are mentioned (Lines 491-494), it would benefit from a clearer subsection or paragraph that includes:

Small sample sizes for team sports

Model instability across random partitions

Generalisation limitations to youth or non-German athletes

6. Conclusion

Line 511 onward: The conclusion is strong but consider softening claims slightly to reflect the proof-of-concept nature of the study. For example:

“These preliminary findings suggest that FFTrees may offer a practical, transparent method for guiding individualised training decisions, though further research with larger samples is recommended.”

7. Language and style

Orthographic errors:

Line 101: “and whether they can be validated…” is incomplete or has a clause error.

Line 249, 304, and several others: Consider reviewing sentence punctuation. Several sentences are long and lack commas, which makes them harder to follow, this may stem from AI-assisted generation.

Recommendation: A thorough proofreading by a native English speaker or professional editor would improve clarity and flow and meet the PLOS ONE criterion for “standard English.”

Final recommendation: major revision

This is a promising and well-structured study with a novel approach, but it would benefit from clearer articulation of objectives, tighter theoretical framing, enhanced methodological transparency, and improvements to writing and formatting.

I encourage the authors to consider the suggestions outlined above, not as criticisms, but as constructive steps toward strengthening their valuable contribution to performance science.

Reviewer #3: Thank you for submitting the manuscript to Plus One.

The topic of the study is highly valuable and, more importantly, practical and experience-based. The authors’ effort to contribute to the enhancement of elite athlete performance by applying evidence-based sport science principles is both commendable and timely.

The significance of the study's outcomes is further reinforced by the choice of participants—German national team athletes—who represent one of the leading nations in both individual and team-based sports at the international level.

I congratulate the research team on their efforts. However, I believe that the following points—if addressed—can substantially improve the quality and impact of the manuscript:

1. The title is somewhat generic. Given the strong applied dimension of this study, I suggest emphasizing the performance enhancement aspect in the title to increase its attractiveness and specificity.

2. Please consider rewriting the final lines of the abstract to emphasize the contribution to performance science and coaching practice more clearly.

3. Although the methodological approach is sound, the sample sizes (n = 27 for trampoline, n = 53 for volleyball) are relatively small. This particularly limits generalizability for the trampoline model and should be addressed more explicitly in the limitations section.

4. What specific challenges did the research team face while athletes completed questionnaires or diagnostic tests, given the constraints of elite training environments? Clarifying this in the discussion or limitations section may be insightful.

5. Clearly identify the cognitive cues used for each sport to help readers better understand the sport-specific adaptations of the FFTrees.

6. Consider simplifying or contextualizing the key outputs (e.g., cut-off values, cue combinations) in a way that is easier for coaches at various levels to interpret and apply.

7. It would enhance the practical relevance of the study to provide concrete, real-world examples of how each cue can be implemented in athlete monitoring or decision-making.

8. Several sentences in the discussion are long and dense. Revising into shorter, clearer statements would improve readability.

9. Please elaborate more on the difference in FFTree performance between individual (trampoline) and team sports (volleyball). This is an important insight for readers.

10. Correct the term "instable FFTrees" to "unstable FFTrees" for better clarity and technical accuracy.

11. If possible, discuss the transferability of this approach to other sports or organizational settings to broaden its application.

6. PLOS authors have the option to publish the peer review history of their article (what does this mean? ). If published, this will include your full peer review and any attached files.

**Do you want your identity to be public for this peer review?** For information about this choice, including consent withdrawal, please see our Privacy Policy .

Reviewer #1: No

Reviewer #2: No

Reviewer #3: **Yes: ** Dr Mahdi Gharibzadeh

---

## [Author Response · Author response to Decision Letter 1]

9 Jul 2025

Dear Editors and Reviewers, thank you for allowing us to submit a revised draft of our manuscript. We appreciate you and the reviewers for your precious time in reviewing our paper and providing valuable comments. It was your insightful feedback that led to improvements in the renewed version. We fully and thoroughly addressed your concerns. Please refer to the file "Response to Reviewers," in which we elaborate on your comments in a point-to-point manner and presented consecutive changes to the manuscript. We hope that, after careful revisions, the manuscript meets your standards. The authors welcome further constructive feedback if any.

We have revised the "financial disclosure" and "data availability statement", and re-uplaoded the affected Supporting Information files. The amended Role of Founder statement is included in the updated cover letter. Thank you for your constructive feedback and for making this adjustment on our behalf.

Dear Reviewer #1, we would like to sincerely thank you for your insightful and encouraging review of our manuscript. We greatly appreciate the time and effort you dedicated to carefully evaluating our paper. We are truly excited and grateful that you recognize the importance and value of the research topic we have addressed. Your positive assessment and summary affirm the relevance of our work and reinforced our enthusiasm for pursuing this research. We have carefully considered your comments as we finalized the revised manuscript.

Dear Reviewer #2, we would like to thank you sincerely for your thorough and detailed review of our manuscript. We are grateful for the careful attention and effort you invested in reviewing our paper. Your thoughtful and detailed feedback, including the helpful examples, and practical suggestions for improvement, has been incredibly valuable to us. Your constructive feedback has inspired us to refine and strengthen the manuscript further. We carefully considered all of your recommendations while revising the manuscript.

Dear Reviewer #3, we are sincerely grateful for your thoughtful evaluation of our manuscript. We deeply appreciate the time and expertise you dedicated to review our paper. Your insightful feedback and practical suggestions for improvement, have been incredibly valuable to us. Your observations have played a significant role in strengthening our manuscript further. We carefully considered all your recommendations while revising the manuscript.

Sincerely

Lena Siebert and coauthors

---

## [Decision Letter · Decision Letter 1]

16 Jul 2025

How fast-and-frugal trees can inform diagnostic and intervention decisions for enhancing elite athlete performance

PONE-D-25-17971R1

Dear Dr. Lena Siebert,

We’re pleased to inform you that your manuscript has been judged scientifically suitable for publication and will be formally accepted for publication once it meets all outstanding technical requirements.

Kind regards,

Mário Espada, PhD

Academic Editor

PLOS ONE

Reviewers' comments:

Reviewer's Responses to Questions

**Comments to the Author**

1. If the authors have adequately addressed your comments raised in a previous round of review and you feel that this manuscript is now acceptable for publication, you may indicate that here to bypass the “Comments to the Author” section, enter your conflict of interest statement in the “Confidential to Editor” section, and submit your "Accept" recommendation.

Reviewer #2: All comments have been addressed

Reviewer #3: All comments have been addressed

2. Is the manuscript technically sound, and do the data support the conclusions?

Reviewer #2: Yes

Reviewer #3: Yes

3. Has the statistical analysis been performed appropriately and rigorously? 

Reviewer #2: Yes

Reviewer #3: Yes

4. Have the authors made all data underlying the findings in their manuscript fully available?

Reviewer #2: Yes

Reviewer #3: Yes

5. Is the manuscript presented in an intelligible fashion and written in standard English?

Reviewer #2: Yes

Reviewer #3: Yes

6. Review Comments to the Author

Reviewer #2: The manuscript is well-conceived, methodologically sound, and clearly written. The objectives, procedures, and conclusions are coherent and contribute meaningfully to the field. No ethical concerns were identified. I am satisfied with the overall quality of the work and recommend it for publication with no major changes.

Reviewer #3: The revisions made to the manuscript have been reviewed and are deemed acceptable. However, it is important to note that all comments provided by the reviewers must be thoroughly addressed in the authors’ response letter. The responses should specifically refer to each comment with detailed explanations. In this case, the responses were presented in a general manner and lacked the necessary completeness.

---

## [Editor Report · Acceptance letter]

PONE-D-25-17971R1

PLOS ONE

Dear Dr. Siebert,

I'm pleased to inform you that your manuscript has been deemed suitable for publication in PLOS ONE. Congratulations! Your manuscript is now being handed over to our production team.

Kind regards,

on behalf of

Dr. Mário Espada

Academic Editor

PLOS ONE